# Between Innate and Adaptive Immune Responses: NKG2A, NKG2C, and CD8^+^ T Cell Recognition of HLA-E Restricted Self-Peptides Acquired in the Absence of HLA-Ia

**DOI:** 10.3390/ijms20061454

**Published:** 2019-03-22

**Authors:** Wiebke C. Pump, Thomas Kraemer, Trevor Huyton, Gia-Gia T. Hò, Rainer Blasczyk, Christina Bade-Doeding

**Affiliations:** 1Institute for Transfusion Medicine, Hannover Medical School, Medical Park, Feodor-Lynen-Str. 5, 30625 Hannover, Germany; Abels.Wiebke@mh-hannover.de (W.C.P.); th.kraemer1@gmail.com (T.K.); Ho.Gia-Gia@mh-hannover.de (G.-G.T.H.); blasczyk.rainer@mh-hannover.de (R.B.); 2Department of Cellular Logistics, Max Planck Institute for Biophysical Chemistry, Am Fassberg 11, 37077 Göttingen, Germany; trevor.huyton@mpibpc.mpg.de

**Keywords:** Immune Evasion hCMV, NKG2A/CD94, NKG2C/CD94, HLA-E peptide ligands

## Abstract

On healthy cells the non-classical HLA class Ib molecule HLA-E displays the cognate ligand for the NK cell receptor NKG2A/CD94 when bound to HLA class I signal peptide sequences. In a pathogenic situation when HLA class I is absent, HLA-E is bound to a diverse set of peptides and enables the stimulatory NKG2C/CD94 receptor to bind. The activation of CD8^+^ T cells by certain p:HLA-E complexes illustrates the dual role of this low polymorphic HLA molecule in innate and adaptive immunity. Recent studies revealed a shift in the HLA-E peptide repertoire in cells with defects in the peptide loading complex machinery. We recently showed that HLA-E presents a highly diverse set of peptides in the absence of HLA class Ia and revealed a non-protective feature against NK cell cytotoxicity mediated by these peptides. In the present study we have evaluated the molecular basis for the impaired NK cell inhibition by these peptides and determined the cell surface stability of individual p:HLA-E complexes and their binding efficiency to soluble NKG2A/CD94 or NKG2C/CD94 receptors. Additionally, we analyzed the recognition of these p:HLA-E epitopes by CD8^+^ T cells. We show that non-canonical peptides provide stable cell surface expression of HLA-E, and these p:HLA-E complexes still bind to NKG2/CD94 receptors in a peptide-restricted fashion. Furthermore, individual p:HLA-E complexes elicit activation of CD8^+^ T cells with an effector memory phenotype. These novel HLA-E epitopes provide new implications for therapies targeting cells with abnormal HLA class I expression.

## 1. Introduction

The discrimination of self and non-self by the immune system is based on the recognition of specific ligands by their cognate immune receptors. This interplay relies on the overall strength of binding to reach a threshold for intracellular signalling, providing either stimulatory or inhibitory action [1]. The identification of self and non-self is possible due to the presentation of antigenic peptide pools on the cell surface of all nucleated cells by HLA class I molecules to be monitored by natural killer (NK) or CD8^+^ T cells [2,3,4].

The protective feature of human leucocyte antigen (HLA) class I antigen presentation is nevertheless susceptible to immune escape mechanisms of certain malignancies or viruses such as human cytomegalovirus (hCMV) and others [5,6,7,8,9,10,11]. Here, the surface expression of HLA class I molecules is impaired by several types of down-regulatory mechanisms mediated for instance during a hCMV infection where HLA class I heavy chains are relocated to proteasomal degradation by viral glycoproteins (gp)US2 or gpUS11 [12,13]. The absence of HLA class I molecules on the cell surface makes the cell prone to a surveillance mechanism executed by NK cells that patrol for HLA class I expression and attack cells that lack HLA class I molecules on their surface, a mechanism also known as “missing self” [14]. Viruses such as hCMV evolved a strategy to evade NK cell killing by down-regulation of HLA class I molecules. This strategy is mediated by the virus-derived protein gpUL40 that provides a peptide ligand to induce surface expression of the HLA class Ib molecule HLA-E. HLA-E surface expression is not affected by the HLA class I down-regulatory mechanisms caused by hCMV immune evasion proteins [15,16]. HLA-E predominantly presents peptides derived from HLA class I signal peptides and is the ligand for the C-type lectin NKG2/CD94 receptor family on NK cells. Regular HLA class I expression can therefore be monitored by NK cells [17]. In particular HLA-E is recognized by either the inhibitory NKG2A/CD94 or the stimulatory NKG2C/CD94 receptor and, therefore, plays a critical role in the balance between cell protection and attack [17,18,19]. This balance is based on a framework of different inhibiting and stimulatory receptors like the killer cell immunoglobulin-like receptors (KIR), NKG2D, NKp46 or NKp30 [20,21,22]. Since HLA-E belongs to the family of HLA class Ib molecules, its variants are mainly restricted to two functional alleles, HLA-E*01:01 and HLA-E*01:03, distinguished by a single amino acid (AA) substitution at position 107 with an Arginine (HLA-E*01:01) to Glycine (HLA-E*01:03) located outside of the peptide binding region [23]. In consequence of its restricted polymorphisms, the immunological role of HLA-E seems to be dependent on the presented peptide that determines the fate of the cell as an outcome of effector cell activation or inhibition. In the presence of HLA class I molecules, HLA-E is reported to present nonameric peptides derived from HLA class I signal peptides, or peptides derived from pathogenic origin [24,25,26,27]. The acquisition of peptide ligands by HLA class I molecules is facilitated by the peptide loading complex in the endoplasmic reticulum (ER). Here, the functionality of major components such as Tapasin or the transporter associated with antigen processing (TAP) is required to maintain regular assembly of peptide:HLA (p:HLA) molecules [28]. A recent study identified over 500 peptide ligands that were presented by HLA-E in the absence of TAP [29]. These peptides showed diverse sequences, but none of the peptides was derived from an HLA class signal sequence.

Our group identified 36 HLA-E peptide ligands that are presented in the absence of HLA class I molecules in LCL 721.221 cells [30] that possess all components of a functional peptide loading complex [31]. These peptides originated from different cellular proteins and display non-canonical features with up to 16 AAs in length and pronounced sequence variability highly divergent from HLA class I derived signal peptides. Additionally, a selection of these peptides revealed an impaired cell protective functionality when bound to HLA-E on artificial antigen presenting cells (APCs). Co-incubation with NK cells resulted in high NK cell mediated cytotoxicity. These data show the variability of HLA-E immune modulation through the presentation of diverse peptides, depending on the individual case of cell abnormality. Hence, we aimed to address the question of whether the lack of cell protection is due to an unstable interaction of p:HLA-E complexes with its cognate NKG2A/CD94 NK cell receptor. Therefore, the binding stability of these p:HLA-E complexes to either stimulatory NKG2C/CD94 or inhibitory NKG2A/CD94 receptors was analyzed in vitro. Furthermore, unusual conformations of these peptides, when bound to the peptide binding region of HLA-E, have been predicted by molecular modelling. The previously reported existence of HLA-E restricted CD8^+^ T cells [25,32,33,34] led us to the assessment if these p:HLA-E complexes might be instead recognized by polyclonal CD8^+^ T cells. The HLA-E restricted self-peptides investigated in this study were obtained from an HLA class I deficient cell line mimicking HLA class I down-regulation. The knowledge and functional analysis of distinct HLA-E restricted peptides that are acquired in the absence of HLA class I signal peptides will provide new implications for hematopoietic cell malignancies and infections where HLA-E expression is still not well understood.

## 2. Results

### 2.1. Non Canonical Peptides Provide Stable HLA-E Cell Surface Expression

We tested four non-canonical HLA-E peptide ligands for their ability to sustain HLA-E surface expression (Table 1). We performed peptide stabilization assays on T2E cells (Appendix A) and the stability of cell surface p:HLA-E complexes was monitored with inhibition of intracellular protein transport by Brefeldin A (BrefA) that provides specific decay of p:HLA-E complexes on the cell surface at the initial time point. The p:HLA-E complex stability was determined without inhibition of intracellular protein transport that reflects the physiological situation. The stability is expressed as DT_50_ (in hours), representing the incubation time for 50% of the complexes to decay (Table 2). The MFI at 0 h was measured and analyzed, after which time the mean fluorescence intensity (MFI) dropped to 50%. All peptides had a DT_50_ > 4 h from which the 13-mer peptide DVHDGKVVSTHEQ (DQ13) showed the highest overall stability after 6 h with only 1.5% reduction in surface expression compared to the VIL9 peptide with 23% loss of surface expression (Figure 1a–e). The two 15-meric peptides showed clear differences in their capability to sustain p:HLA-E complexes with a DT_50_ of < 6 h with LVDSGAQVSVVHPNL (LNL15) and LGHPDTLNQGEFKEL (LEL15) with a DT_50_ of > 6 h. The 10-mer SKGKIYPVGY (SY10) demonstrated a DT_50_ of > 6 h and belonged to the group of good stabilizing peptides. Stability of cell surface p:HLA-E was higher or at least the same level for all tested peptides in the absence of BrefA with DT_50_ values > 6 h.

### 2.2. Distinct Non-Canonical HLA-E Peptides Impact the Binding to sNKG2A/CD94 or sNKG2C/CD94 Heterodimers

To determine if the individual p:HLA-E complexes show differences in binding to sNKG2A/CD94 or sNKG2C/CD94 receptors in vitro, we produced soluble forms of their extracellular domains. The initial p:HLA-E cell surface levels showed no relevant differences after the peptide stabilization assay on T2E cells that excludes a correlation between differences in NKG2/CD94 receptor binding levels and available initial p:HLA-E molecules on the cell surface (Figure 2a). By comparing the inhibitory NKG2A/CD94 and the activating NKG2C/CD94 receptors, no significant differences in binding to any of the individual p:HLA-E molecules could be observed after 2 h of incubation (Figure 2b). However, the binding of sNKG2A/CD94 or sNKG2C/CD94 illustrated variability and was dependent on the p:HLA-E complex studied. The binding of sNKG2A/CD94 or sNKG2C/CD94 to HLA-E bound to the LNL15 peptide was highly reduced in comparison to HLA-E bound to the DQ13 or VIL9 peptides. While this difference was only significant for sNKG2A/CD94, the sNKG2C/CD94 receptor showed similar trends. The p:HLA-E complexes with the LEL15 or SY10 peptides showed a tendency toward higher stability with the sNKG2A/CD94 and sNKG2C/CD94 receptors than the LNL15 peptide; however, this was not as high as p:HLA-E with DQ13 or VIL9.

### 2.3. Non-Canonical HLA-E Peptides Induce HLA-E Restricted CD8^+^ T Cell Proliferation

To highlight the role of the distinct p:HLA-E complexes in adaptive immune responses, we analyzed the p:HLA-E recognition by CD8^+^ T cells. The analyzed peptides were derived from HLA-E molecules in the absence of HLA class I molecules that artificially mimic the situation during viral immune evasion; e.g., by hCMV. All test peptides were examined for their capacity to induce CD8^+^ T cell proliferation determined by carboxyfluorescein succinimidyl ester (CFSE) dilution (Figure 3). Proliferation serves as a first marker for p:HLA-E recognition by T cells. To demonstrate that proliferation is exclusively induced by p:HLA-E complexes, we used T2E cells loaded with the test peptides as APCs and co-cultured them with purified CD8^+^ T cells from PBMCs. For proliferation analysis, cells were gated on CD3^+^CD8^+^ cells. Proliferation was considered as specific after subtracting the percentage of proliferated CD8^+^ T cells co-cultured with the T2E control. Samples with 10% specific proliferation or more were considered positive. CD8^+^ T cells from both donors showed a strong proliferation induced by three out of the five tested peptides with DQ13, LNL15, and LEL15 (Table 2). The remaining HLA-E bound peptides did not induce any specific proliferation. Taken together, the results indicate that CD8^+^ T cells recognizing the HLA-E epitopes DQ13, LNL15, and LEL15 are detectable with high frequencies, indicating a high immunogenicity for these three p:HLA-E complexes.

### 2.4. HLA-E induced CD8^+^ T Cells Show an Effector Phenotype and Low Induction of Natural Killer Cell Receptors Expression

To determine if the respective proliferated CD8^+^ T cell population shows a shift from naïve state into effector memory cells, we determined the surface expression of CD45RA and CD45RO before and after stimulation with T2E cells. The stimulation of CD8^+^ T cells with distinct p:HLA-E complexes resulted in the loss of CD45RA^+^ cells that represent naïve T cell populations and the gain of CD45RO^+^ expression on CD8^+^ T cells that were encountered, with T2E cells presenting the DQ13, LNL15 or LEL15 peptide (Figure 4a). The expression of the CD45RO effector memory marker is in line with the strong T cell proliferation response that was induced by these peptides. CD8^+^ T cells stimulated with the SY10 or VIL9 peptide showed no shift in CD45RO expression in comparison to CD8^+^ T cells that were co-incubated with T2E cells without peptide. The effect of HLA-E antigen presentation on the cell surface expression of NK cell receptors, in particular the NKG2A/CD94 or NKG2C/CD94 receptor, on the examined CD8^+^ T cell population was analyzed to determine if there is a correlation between T cell activation and NK cell receptor expression induced by HLA-E (Figure 4b). The surface levels of NKG2C/CD94 on CD8^+^ T cells derived from donor #1 showed a highly positive subpopulation with 62% for this receptor before stimulation with T2E cells, whereas CD8^+^ T cells derived from donor #2 contained only around 8% of a subpopulation positive for the NKG2C/CD94 receptor. However, when we examined HLA-E complexes bound to DQ13, LNL15, or LEL15 that induced proliferation and expression of CD45RO, an almost complete loss of this NKG2C/CD94^+^CD8^+^ T cell subpopulation within the CD8^+^ T cells derived from the tested donors was observed. Regarding the NKG2A/CD94^+^ subpopulations of CD8^+^ T cells, no distinct shift could be observed compared to pre-stimulation.

## 3. Discussion

The balance between stimulatory and inhibitory signals elicited from the NKG2C/CD94 and NKG2A/CD94 receptor is highly dependent on the peptide that is presented by their common ligand HLA-E [36]. We recently reported the presentation of non-canonical peptides by HLA-E. These peptides impact the recognition of HLA-E on APCs by monitoring NK cells that subsequently led to NK cell mediated lysis, suggesting a loss of cell protective recognition by the cognate NKG2A/CD94 inhibitory receptor [30]. To determine how distinct p:HLA-E complexes influence the binding to the NKG2/CD94 receptors, we determined the differences in binding to recombinant sNKG2A/CD94, or sNKG2C/CD94 receptors consisting of their soluble ectodomains at physiological conditions. The data presented here demonstrates that HLA-E molecules bound to extraordinary long peptides of up to 15 AAs in length are still able to bind to NKG2/CD94 receptors at almost equal levels. We could show that these p:HLA-E complexes bind to sNKG2A/CD94 or sNKG2C/CD94 heterodimers at physiological conditions in vitro. The data suggest a fine balance of inhibitory and stimulatory signals resulting from receptor recognition; however, the in vivo significance is unclear since the availability of receptors on the cell surface of NK cells and the signaling threshold that is required for engagement is unknown. We could show that sNKG2/CD94 binding is dependent on the peptide that is bound to HLA-E, resulting in highly decreased binding of both receptors to HLA-E bound to LNL15 compared to VIL9 or DQ13. HLA-E bound to the VIL9 peptide resulted in the highest binding of the sNKG2A/CD94 receptor, elucidating its role as a protective p:HLA-E complex against NK cell activation. The VIL9 peptide derives from the signal sequence of HLA-C and is a natural ligand for HLA-E. Moreover, the same sequence exists in the gpUL40 from hCMV to restore HLA-E surface expression in hCMV infected cells to escape NK cell cytotoxicity. One explanation for the observed binding to the sNKG2/CD94 receptors in the current study could be that the CD94 part of the heterodimeric complex is responsible for the major interaction with the HLA-E molecule, and that NK cell activation is highly dependent on the additional interaction between residues of the presented peptide and the NKG2A or NKG2C part of the receptor. Structural analysis of HLA-E bound to the signal peptide derived from HLA-G VMAPRTLFL in complex with the NKG2A/CD94 receptor clearly depicted that one AA difference within the peptide determines the HLA-E/receptor affinity and subsequently the cellular fate [37,38]. Furthermore, a six fold difference in binding affinity between certain p:HLA-E/NKG2A/CD94 and p:HLA-E/NKG2C/CD94 complexes determined by surface plasmon resonance experiments has been reported [39,40]. Although these studies provide valuable information about the biophysical behavior of interactions between p:HLA-E complexes and NKG2/CD94 receptors, their non-physiological conditions regarding bacterial protein production, in vitro refolding, and temperature might limit the conclusions for the physiological binding behavior. Moreover, Kaiser et al. [40] showed that even a single AA substitution at position P5 of the HLA-G derived HLA-E restricted nonamer peptide ligand VMAP*R*TLFL to VMAP*K*TLFL led to similar binding affinities of both receptors NKG2A/CD94 and NKG2C/CD94 to this p:HLA-E complex. The conclusion that diverse p:HLA-E complexes generally bind with a six fold lower affinity to NKG2C/CD94 compared to NKG2A/CD94 is rather hard to reach. A direct binding of soluble CD94 alone to HLA class I antigens has been shown before [41], nevertheless the heterodimeric NKG2/CD94 complex is required for the receptors cell surface expression [42]. All peptides in this study provided stable p:HLA-E complexes that could be recognized by the sNKG2/CD94 receptors.

The presentation of non-canonical peptides by HLA-E in the absence of HLA class I molecules and the impaired functionality in NK cell receptor activation suggest that these p:HLA-E complexes might fulfill a dual role. It is known that HLA-E is also recognized by CD8^+^ T cells and can induce cytotoxic T cell responses [25,32,43]. All tested peptides are derived from the self-peptidome of HLA class I negative LCL 721.221 cells, naturally presented by HLA-E molecules, and shown to restore stable cell surface p:HLA-E molecules. The p:HLA-E complexes formed by these peptides were previously studied on their impact on NK cell regulation and displayed an impaired cell protection due to the loss of recognition by the NKG2A/CD94 receptor on NK cells [30]. The predicted HLA-E molecule structures revealed distinct conformations of the individual peptides bound to the peptide binding region of HLA-E that most likely prevent sufficient binding of the NKG2/CD94 heterodimer to induce signaling. Since HLA-E presents canonical HLA class I derived signal peptides physiologically, the interaction with the invariant NKG2/CD94 receptors is rather restricted to p:HLA-E complexes with canonical peptides. The predicted unusual conformations of the test peptides that expose parts of their AAs to solvent in a bulgy like structure suggest a higher dynamic flexibility in comparison to canonical peptides that form a stretched conformation rather buried in the peptide binding groove of HLA-E. The implication of an immunogenic functionality of these p:HLA-E complexes raised the question of if these molecules would be available for T cell receptor (TCR) recognition by CD8^+^ T cells. CD8^+^ T cells from healthy blood donors have been tested for their ability to differentially recognize p:HLA-E epitopes. Taken into account that the peptide repertoire of HLA-E might be different in a clinical setting where hCMV infected cells undergo viral immune evasion, we here describe an artificial immune evasion system to investigate if there is a differential peptide mediated activation of CD8^+^ T cells by novel HLA-E epitopes. The proliferation pattern of CD8^+^ T cells revealed a peptide mediated recognition of HLA-E bound to the DQ13, LNL15 or LEL15 peptide. Since T2 cells express low levels HLA-A2 on their surface [44] that might be recognized by CD8^+^ T cells, a proliferation response of CD8^+^ T cells co-incubated with T2E without addition of peptide provided the reference for peptide-specific proliferation. Given the highly limited structure variability of HLA-E, one might have expected a more uniform recognition pattern of the analyzed peptides. It has been shown previously that TCRs recognize p:HLA complexes bound to long peptides (>10 AAs in length) with alternative docking modes [45]. These alternative docking modes are based on exposed residues of the designated peptide and not on TCR binding to residues located at the α1- or α2-helices of the HLA heavy chain. The unusual long peptides tested in our study likely provide unusual conformations that broaden the availability of p:HLA-E complexes for CD8^+^ T cell recognition. In consequence to the recognition of these p:HLA-E complexes by CD8^+^ T cells, the loss of CD45RA and the gain of CD45RO surface expression shows that these epitopes were recognized by the majority of CD8^+^ T cells which in turn differentiated to the memory T cell compartment after antigen activation. However, the CD8^+^ T cells that proliferated in the presence of T2E cells bound to DQ13, LNL15 or LEL15 displayed low expression of the NK cell receptors NKG2A/CD94 or NKG2C/CD94 on their surface; the expression levels were similar to those of the controls. The expression of NKG2/CD94 receptors is a general feature of blood circulating CD8^+^ T cells [46], where increased levels of NKG2A/CD94 on activated CD8^+^ is associated with inhibitory regulation of these CD8^+^ T cells to contribute to the fine balance between the elimination of abnormal cells and healthy tissue [47]. The contribution of NKG2C/CD94 expression on CD8^+^ T cells is not fully understood yet, but is thought to most likely display a co-stimulatory receptor of CD8^+^ effector cells [48]. The NKG2C/CD94 induced proliferation of CD8^+^ T cells correlates with the expression of NKG2C/CD94 on the cell surface of proliferating cells [49], therefore we cannot rule out a NKG2C/CD94 contribution to CD8^+^ T cell activation. As stressed above the low expression of NKG2C/CD94 on a small subset of proliferated CD8^+^ T cells after stimulation might display a co-stimulatory role for an HLA-E- NKG2C/CD94 interaction. However, the majority of proliferated CD8^+^ T cells after stimulation is NKG2C/CD94 negative. Additionally, the SY10 and VIL9 peptides did not induce a proliferation response as could have been expected especially for the latter one as a high affinity HLA-E peptide ligand. These observations strengthen our knowledge of the repertoire of epitopes in HLA-E restricted immune regulation. Furthermore, Oliveira et al. [50] identified peptide ligands of the HLA-E mouse homologue Qa-1^b^ in a TAP-deficient mouse cell line that showed a variability in sequence and length with peptides consisting of up to 18 AAs and no peptide binding motif. However, these peptides elicited a strong cytotoxic T cell response that underlines the capability of Qa-1^b^ and HLA-E to present novel epitopes for T cell recognition.

Our study shows that HLA-E presents stable epitopes that are comparable to p:HLA-E complexes bound to peptides derived from HLA class I molecules (e.g., the 9-mer VIL9). Furthermore, the peptide mediated recognition of HLA-E by CD8^+^ T cells could guide towards the development of new cellular therapeutic strategies. These HLA-E restricted CD8^+^ T cells reveal a CD45RA^-^CD45RO^+^ phenotype. Our data reports novel HLA-E presented antigens that demonstrate HLA-E’s capacity to form likely unique epitopes that are probably frequently presented on cells with abnormal or no HLA class I expression or with a defective peptide processing machinery as shown in mice [50,51]. The recognized p:HLA-E complexes with DQ13, LNL15 or LEL15 by CD8^+^ T cells from distinct individuals illustrates that HLA-E potentially plays a more diversified role in immunity than previously thought. However, these particular peptides suggest a novel functionality of HLA-E, but presumably do not reflect specific therapeutic peptides since they are cell line derived. Nevertheless, the frequency of these kinds of unusual p:HLA-E complexes on affected cells might not be as high as with HLA class I derived peptides. The observed CD8^+^ T cell proliferation induced by these p:HLA-E complexes demonstratively shows that HLA-E bound to non-canonical peptides could serve as a source of informative molecules on diseased cells. These results suggest the existence of a potent recognition mode of cognate TCRs on CD8^+^ T cells. Obviously, there is a sustained need for the analysis and understanding of HLA-E restricted peptides in cells that are affected by tumor or virus induced impaired HLA class I presentation machinery. Thus, HLA-E epitopes displaying non-canonical peptides could serve as a biomarker to identify abnormal cells that could complement adoptive effector cell therapies. Therefore, the identification of physiopathological HLA-E peptide ligands and development of single high affinity receptors such that have been shown for chimeric antigen receptors on T cells against acute lymphoid leukemia [52] is one obstacle that needs to be overcome for future therapy concepts.

## 4. Materials and Methods

### 4.1. Cells

The human HLA negative leukemia cell line K562 and the TAP deficient cell line T2 were maintained in RPMI 1640 (Life Technologies, Darmstadt, Germany) supplemented with 10% heat inactivated fetal calf serum (FCS), 2 mM L-glutamine, 100 U/mL penicillin and 100 µg/mL streptomycin (c.c.pro, Oberdorla, Germany). For production of lentiviral particles, HEK293T cells were cultured in DMEM (Life Technologies) supplemented with 10% heat inactivated FCS, 2 mM L-glutamine, 100 U/mL penicillin, 100 µg/mL streptomycin and 1 mg/mL Geneticin (Life Technologies). Culture conditions were equal for all cells (37 °C, 5% CO_2_). PBMCs were obtained from healthy blood donors that had signed informed consent (Hannover Medical School).

### 4.2. Peptides

All tested peptides were synthesized and purchased from Thermo Fisher Scientific (Ulm, Germany) and dissolved in DMSO (100 mg/mL).

### 4.3. Construction of Vectors Encoding for Soluble NKG2/CD94 Heterodimers

The plasmids containing the extracellular domain of human NKG2A (AA residues 99–233), NKG2C (residues 98–231) and CD94 (residues 34–179) were synthesized and purchased from Life Technologies (Darmstadt, Germany). One construct contained either the NKG2A or NKG2C domain and the CD94 domain, both separated by a T2A sequence [53] that leads to equal expression of soluble (s)NKG2 and sCD94 as both proteins form one open reading frame and are separated during translation. Each construct also possesses a murine Ig κ-chain leader sequence that allows secretion of the proteins into the cell culture medium [54]. Additionally, a V5 peptide tag [55] is encoded downstream of the CD94 domain in each construct. DNA fragments encoding the entire Igκ/sNGK2/T2A/Igκ/sCD94/V5 construct were cloned into the lentiviral pRRL.PPT.SFFV.mcs.pre vector at the *BamHI* and *XbaI* sites.

### 4.4. Production of Soluble NKG2/CD94 Heterodimers

To produce lentiviral particles for stable transduction of K562 cells with the above mentioned receptor constructs, HEK293T cells were separately transfected with the lentiviral pRRL/sNKG2A/CD94 or pRRL/sNKG2C/CD94 vectors as previously described [30]. K562 cells were then lentivirally transduced with the respective viral particles. sNKG2A/CD94 or sNKG2/CD94 producing K562 cells were cultured in bioreactors (CELLine, Integra, Fernwald, Germany), and the supernatant was harvested weekly. The soluble heterodimers were directly purified from the cell culture supernatant by affinity chromatography with an N-hydroxysuccimide (NHS)-activated HiTrap column (GE Healthcare, Munich, Germany) coupled with an anti-hNKG2A (clone 121411, R&D systems, Minneapolis, MN, USA) or anti-hNKG2C antibody (clone 134522, R&D systems) that recognizes only the receptor heterodimers. The purified proteins were concentrated by size exclusion with centrifugal Amicon^®^ Ultra filters (Merck Millipore, Schwalbach, Germany) with a cut off of 10 kDa. Concentrated protein was collected from the filter in 75 mM Tris buffer (pH 8.0). Since the elution after affinity chromatography was performed with a 0.1 M glycine/HCl buffer (pH 2.7), the purity of heterodimers in the eluted fraction was proven by native PAGE (Appendix A).

### 4.5. Stabilization Assay of HLA-E Surface Molecules and Detection of sNKG2/CD94 Binding

For stabilization of HLA-E molecules on the cell surface in peptide binding assays T2 cells were transduced with HLA-E*01:01 (T2E) as described previously [30]. 2.5 × 10^5^ T2E cells were incubated with the respective peptide at a concentration of 300 µM for 18 h hours at 37 °C in serum free medium (RPMI 1640) (Appendix A). HLA-E surface expression was detected with the PE-labeled anti-HLA-E mAb 3D12 (BioLegend, San Diego, USA) by flow cytometry (FACS Canto II, BD Biosciences, Heidelberg, Germany). T2E cells without a peptide served as negative control, HLA-E bound to the VMAPRTLIL peptide (200 µM) that is reported to be an HLA-E peptide ligand [35] as positive control. HLA-E cell surface levels were expressed as the geometric mean fluorescence intensities (gMFI) and the MFI was calculated by the formula:MFI = gMFI_sample_ − gMFI_negative control_

The stability of individual p:HLA-E complexes was monitored as previously described [56]. Briefly, over a time period of 6 h at 37 °C, HLA-E surface levels were obtained every 2 h. Decrease in HLA-E levels on the cell surface is considered as decay of p:HLA-E complexes by dissociation of the peptide from the HLA-E molecule. After peptide incubation for 18 h at 37 °C the cells were either washed two times with PBS to remove free peptide and subsequently resuspended in 200 µL serum free RPMI medium or were incubated for 1 h at 37 °C with BrefA at a final concentration of 10 µg/mL. Incubation with BrefA leads to inhibition of transport of *de novo* synthesized HLA molecules from the ER to the cell surface [57]. After BrefA incubation the cells were washed and resuspended in medium as mentioned above. For the initial time point (t = 0 h) HLA-E surface levels were determined immediately. The cells were then further incubated at 37 °C, and HLA-E surface levels were determined after 2, 4 and 6 h after the initial time point.

To test whether the recombinant NKG2/CD94 proteins bind these p:HLA-E complexes and whether differences between sNKG2A/CD94 and sNKG2C/CD94 binding can be observed, flow cytometric analyses with peptide loaded T2E cells were performed. For all tests, 1 × 10^6^ peptide loaded T2E cells were incubated with 200 nM of purified sNKG2A/CD94 or sNKG2C/CD94 receptors in 100 µL serum free RPMI medium. The receptor concentration was titrated for activity with T2E cells loaded with the VMAPRTLIL (VIL9) reference peptide. T2E cells without a peptide served as negative control. After 2 h of incubation at 37 °C, the cells were incubated for 30 min at 4 °C with the anti-V5-tag antibody, the goat-anti-mouse PE-coupled secondary antibody (BD Biosciences) was used for detection (Appendix A). Determination of differences in HLA-E cell surface levels and binding of sNKG2A/CD94 or sNKG2C/CD94 was performed by one-way ANOVA analysis followed by Newman-Keuls post-hoc test.

### 4.6. Proliferation Analysis of CD8^+^ T cells

CD8^+^ T cells were purified from PBMCs by negative selection using magnetic beads, resulting in depletion of CD4^+^, CD14^+^, CD16^+^, CD19^+^, CD20^+^, CD36^+^, CD56^+^, CD66b^+^, CD123^+^, TCRγ/δ^+^ and glycophorin A^+^ cells (Human CD8^+^ T Cell Enrichment Kit, Stemcell Technologies, Vancouver, BC, Canada). Purified CD8^+^ T cells were labeled with CFSE (2 µM, Life Technologies, Darmstadt, Germany), the CD8^-^ fraction was irradiated at 30 Gy and added to the culture as feeder cells. Specific p:HLA-E presenting APCs were obtained from incubation of T2E cells with individual peptides (300 µM) for 18 h at 37 °C. Peptide loaded target cells were irradiated at 40 Gy. 2 × 10^5^ CFSE labeled CD8^+^ T cells were co-cultured with 1 × 10^5^ peptide loaded T2E cells in the presence of 5 × 10^5^ CD8^−^ feeder cells in 96 well roundbottom plates. Cells were cultured in RPMI supplemented with 10% human serum (C-C-Prom, Oberdorla, Germany) and 5 ng/mL IL-7 (Peprotech, NJ, USA). On day 7 of co-culture, IL-2 (Peprotech) was added to a final concentration of 10 U/mL. CD8^+^ T cells co-incubated with T2E cells without peptide or with medium only were carried along as controls. On day 10 of co-culture, cells were harvested and stained with CD3-PE-Cy7 and CD8-PerCP (both from BD Bioscience, Heidelberg, Germany). Cells were gated on CD3^+^CD8^+^ and CFSE dilution was analyzed (Appendix A).

### 4.7. Flow Cytometry Analysis

After 10 days of co-culture, the CD8^+^ T cells were phenotypically analyzed by flow cytometry. Cells were stained with anti- CD3-PE-Cy7, CD8-PerCP, CD56-FITC, CD45RA-APC-H7 and CD45RO-PE-Cy7 mAb (all from BD Biosciences). Furthermore, we used anti-NKG2A-APC and anti-NKG2C-PE mAb (R&D systems, Minneapolis, MN, USA). Live/Dead discrimination was performed by 7-AAD staining (Biolegend, San Diego, CA, USA) according to manufacturer protocol (Appendix A). All samples were analyzed on a FACS Canto II (BD Biosciences).

## Figures and Tables

**Figure 1 ijms-20-01454-f001:**
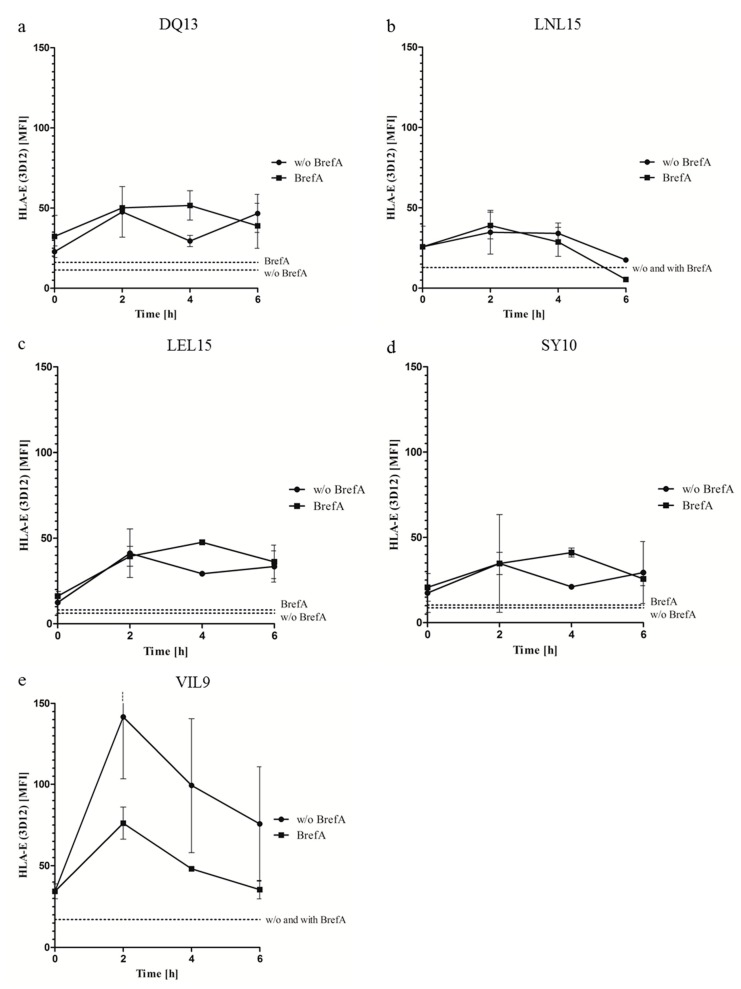
Surface stability of distinct p:HLA-E complexes over a time period of 6 h. T2E cells pre-loaded with one of the five test peptides (**a**–**e**) was incubated with or without BrefA. Binding of the HLA-E specific mAb 3D12 to these cells was analyzed by flow cytometry over 6 h. The depicted values at each time point are the means of three independent experiments ± SEM. The dashed lines indicate the MFI after 50% of the complexes decayed. The crossing of a curve with this line marks the DT_50_ (reported in Table 2). BrefA: Brefeldin A.

**Figure 2 ijms-20-01454-f002:**
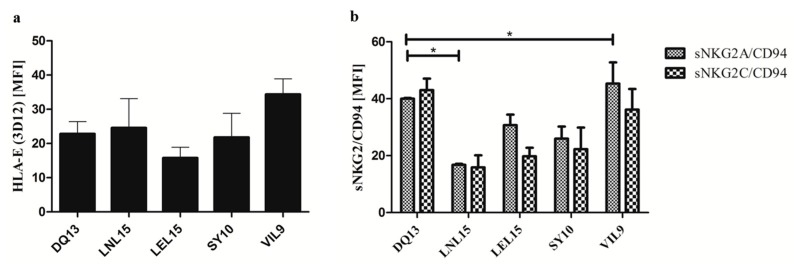
HLA-E surface expression and binding of sNKG2/CD94 receptors to distinct p:HLA-E complexes. (**a**) HLA-E cell surface levels on T2E cells (as MFI = median fluorescence intensity) measured by flow cytometry analysis prior to incubation with either sNKG2A/CD94 or sNKG2C/CD94 receptors. Data is represented as mean ± SEM. Differences in HLA-E expression are non-significant in one-way ANOVA analysis. (**b**) After incubation of peptide pulsed T2E cells with the soluble receptors (sNKG2A/CD94 and sNKG2C/CD94) the levels of receptor binding to the distinct p:HLA-E complexes were determined. Indicated are the means of the MFI of three independent experiments ± SEM. (*) shows significance with *p* < 0.05 using one-way ANOVA analysis and Newman-Keuls post-hoc test for each receptor.

**Figure 3 ijms-20-01454-f003:**
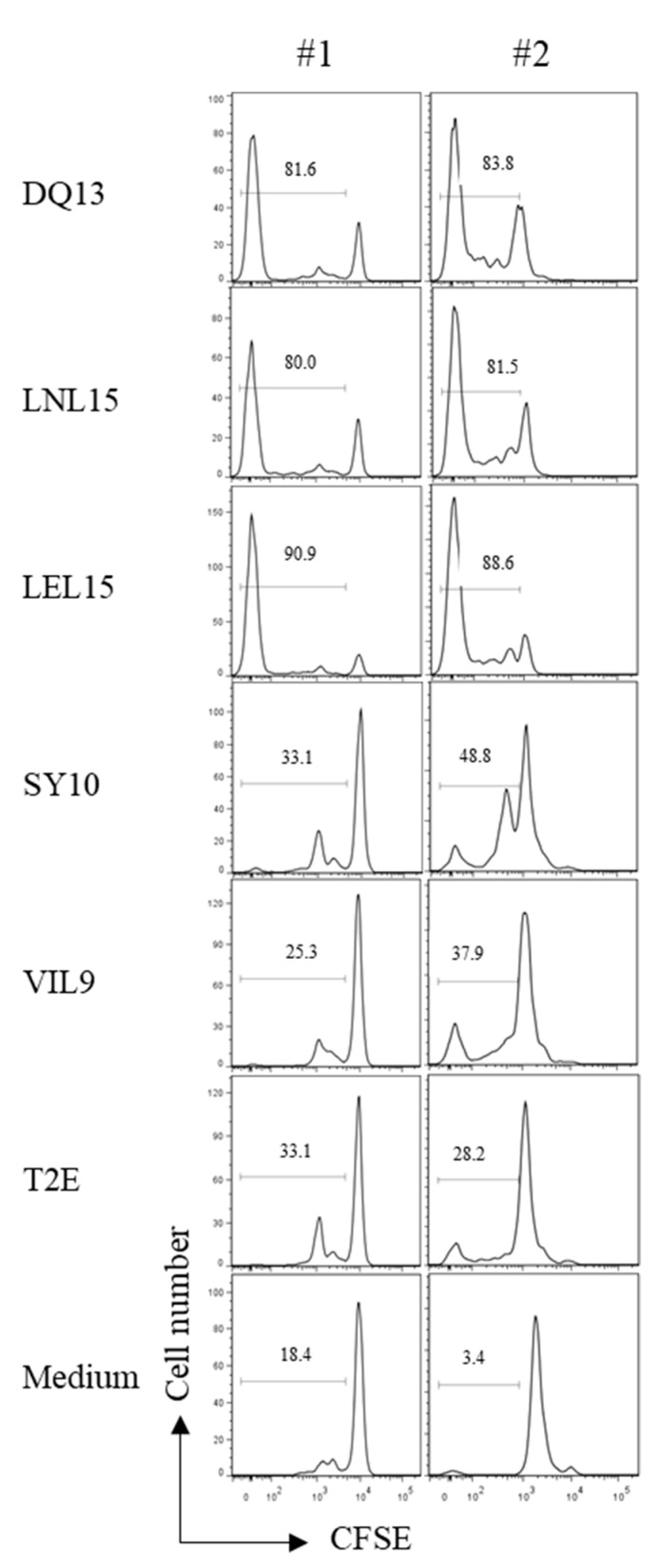
Proliferation profiles of CD8^+^ T cells after stimulation with peptide pulsed T2E cells. CD8^+^ T cells were isolated from PBMCs labeled with CFSE and stimulated with peptide pulsed and irradiated T2E cells to analyze which peptides are capable to activate T cells. T2E cells without peptide (T2E) and medium only were used to determine unspecific proliferation. Histograms are gated on CD3^+^CD8^+^ cells. Depicted numbers in each graph indicate for the percentage of proliferated cells. Shown are results from PBMCs from two different individuals (#1, #2).

**Figure 4 ijms-20-01454-f004:**
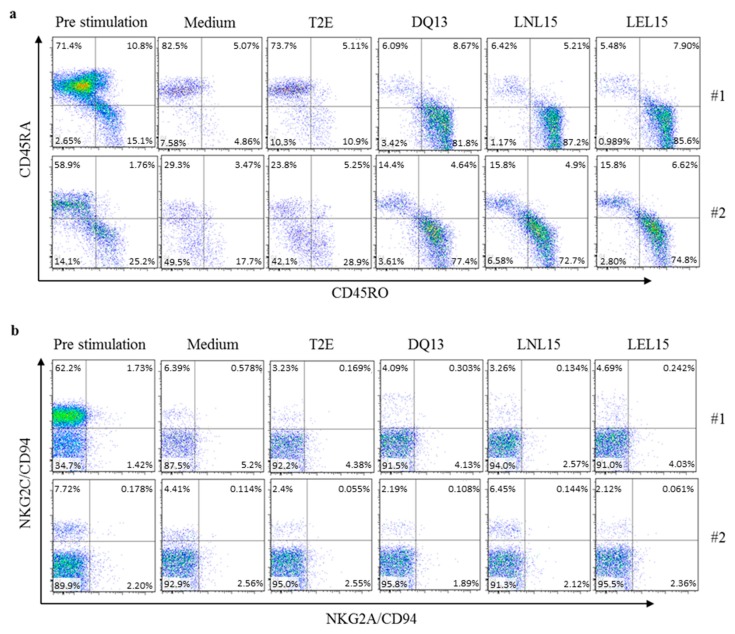
Surface phenotype of HLA-E restricted CD8^+^ T cells. CD8^+^ T cells that proliferated in response to distinct p:HLA-E complexes on T2E cells were assessed for (**a**) the surface expression of CD45RA or CD45RO and (**b**) the expression of NKG2A/CD94 or NKG2C/CD94 receptors after 10 days of stimulation via flow cytometry. Pre stimulation (day 0), incubation with cell medium only and incubation with T2E cells without peptide are displayed as controls. Shown are results from PBMCs from two different individuals (#1, #2).

**Table 1 ijms-20-01454-t001:** Non-canonical HLA-E peptide ligands.

Name	Sequence	Origin	Identified in Reference
DQ13	DVHDGKVVSTHEQ	KRT14 Keratin	[30]
LNL15	LVDSGAQVSVVHPNL	ASPRV1 Retroviral-like aspartic protease 1	[30]
LEL15	LGHPDTLNQGEFKEL	Calprotectin S100A9	[30]
SY10	SKGKIYPVGY	Corneodesmosin	[30]
VIL9	VMAPRTLIL	HLA-C and gpUL40 (hCMV protein)	[35]

**Table 2 ijms-20-01454-t002:** Differences in CD8^+^ T cell proliferation and cell surface stability provoked by p:HLA-E complexes.

p:HLA-E	CD8^+^ T Cell Proliferation(= %p:T2E-%T2E) ^a^		DT_50_ ^c^ [h]
	#1	#2	
DQ13	48.5%	55.6%	> 6
LNL15	46.9%	53.3%	> 4
LEL15	57.8%	60.4%	> 6
SY10	no ^b^	no ^b^	> 6
VIL9	no ^b^	no ^b^	> 6

#1, #2: shown are results from PBMCs from two different individuals. ^a^ T cell proliferation was calculated by subtracting the proliferation rate without peptide (%T2E = control) from the proliferation rate with the respective peptide (%p:T2E); ^b^ no proliferation is defined as proliferation that is ≤10% higher than the proliferation of the control (%T2E); ^c^ DT_50_ represents the incubation time for 50% of the complexes (based on the MFI at 0 h) to decay in the presence of Brefeldin A (presented in Figure 1).

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
