# Peer review of "Between Innate and Adaptive Immune Responses: NKG2A, NKG2C, and CD8+ T Cell Recognition of HLA-E Restricted Self-Peptides Acquired in the Absence of HLA-Ia"

_ijms, 2019, doi:10.3390/ijms20061454_

Round 1
Reviewer 1 Report
In the manuscript entitled ‘Between Innate and Adaptive Immune Responses: HLA-E Restricted Self-Peptides Acquired During an Artificial hCMV Infection Determine the Cell Fate’ the authors demonstrate that self-peptides derived from HLA-class I-deficient cell lines are presented on the cell surface in the context of the non-polymorphic HLA-E molecule. The protein:HLA-E complexes recognize both soluble activating and inhibitory receptors. Finally, the authors also determine that peptide-pulsed cells can be recognized by CD8+ T cells to induce cell proliferation and differentiation into cells expressing effector memory markers. While the manuscript offers intriguing insights into the various potential roles of peptide:HLA-E complexes on the cell surface the data does not present a fully elucidated story. Below are a list of major and minor concerns.
Major concerns
1 – The title of the manuscript overreaches the data in several areas. Firstly, it is incorrect to state that the cell system presented represents an ‘artificial hCMV infection’. It has been well documented that CMV infection massively alters the peptide:HLA-E complexes presented on the cell surface due to a myriad of viral immune evasion mechanisms in addition to downregulation of HLA class I molecules (Hansen et al, 2016 Walters et al, 2018 and others). The title misleads the reader into assuming the data presented reflects what has been observed during CMV infections and no experiments using CMV are performed to validate the artificial system and only a cursory mention of the extensive literature regarding CMV and HLA class I molecules is mentioned in the Introduction and even less so in the Discussion. The manuscript would be better served with a title reflecting the use of a system where HLA class I molecules are absent. Secondly, there is no data presented to indicate the ultimate fate of a cell expressing the particular peptide:HLA-E molecule studied in this manuscript. In fact, Figure 2 indicates that either activation or inhibition of NK cell activity if possible due to roughly equivalent binding of both types of receptors. Activation and proliferation of CD8+ T cells are demonstrated, but the effector function of the stimulated cells is not directly demonstrated.
2 – How is the data presented in Figure 1 different than that previously published by the authors (Kraemer et al, 2015) using the same peptides? In fact, the authors use the same peptides in this publication to show no protection against NK cell cytotoxicity, which would be important to mention in this manuscript when providing the rationale for the experiments presented in Figure 2.
3 – Statistical analysis is not provided for many of the figures. Are there statistics for Figure 2 to support the relative differences in stability discussed by the authors? In the Discussion (lines 205-206) the authors suggest that the VIL9 peptide binds best to the NKG2A/CD94 receptor, but the data as presented does not support this assertion. What do the percentages presented in Table 2 represent – the data from Figure 3?
Minor concerns
1 – An English language editor would be recommended as in several instances it was unclear what the authors were trying to convey.
2 – The scale of Figure 1 makes it difficult to assess the values and percent changes discussed in the text.
3 - What is the significance of the data in Figure 1 +/- Brefeldin A? The authors mention the differences in the Results section but do not convey their interpretation of the results.
4 – The values on the graphs in Figure 4 are too small to read in the print version of the manuscript
Author Response
Reviewer 1:
Major concerns
1. We agree with the reviewer and amended the title for clarity.
We acknowledge the reviewers request to explain the difference between the peptide stabilization experiments that we demonstrated in the paper from 2015 (HLA-E: Presentation of a Broader Peptide Repertoire Impacts the Cellular Immune Response—Implications on HSCT Outcome; Kraemer et al., 2015) and the peptide stabilization experiments in the present manuscript. In the previous paper we determined the saturation concentration of peptides (µM) required to obtain stable pHLA-E complexes for the functional experiments. We measured the stability of pHLA-E complexes at different time points, however we did not acknowledge the development of de novo synthesized HLA-E molecules. In the present paper, we used the previously determined concentration of 300 µM and demonstrate the dissociation of the peptide from the HLA-E molecule, the decay of p:HLA-E complexes, by inhibiting the transport of de novo synthesized HLA molecules from the endoplasmic reticulum to the cell surface with BrefA. Using this method, we could monitor the stability of cell surface p:HLA-E complexes. Furthermore, p:HLA-E complex stability was determined without inhibition of intracellular protein transport that reflects the physiological situation. Here, we determined the stability as DT50 (in hours), representing the incubation time for 50 % of the complexes to decay.
We acknowledge the reviewers request to explain the rationale for the NKG2A/2C receptor binding experiments. We observed in the previous paper (HLA-E: Presentation of a Broader Peptide Repertoire Impacts the Cellular Immune Response—Implications on HSCT Outcome; Kraemer et al., 2015) peptide-specific differences in the NK cell mediated lysis. Those differences could have been elicited through the strength of receptor binding. It has been shown before, that NKG2A and NKG2C can bind to the same pHLA-E complex with different affinities (Kaiser, B.K., et al., Structural basis for NKG2A/CD94 recognition of HLA-E. Proc Natl Acad Sci U S A, 2008). We engineered heterogenic CD94/NKG2A or CD94/NKG2C complexes and detected their binding to the different pHLA-E complexes. In the present manuscript we discussed this rationale (Discussion, page 18, 2nd paragraph; page 19, 1st paragraph). In Table 1, page 9, we refer to the peptide sequences and the respective reference.
2. The stability of cell surface p:HLA-E complexes was monitored with inhibition of intracellular protein transport by BrefA that provides specific decay of p:HLA-E complexes on the cell surface at the initial time point. Furthermore, p:HLA-E complex stability was determined without inhibition of intracellular protein transport that reflects the physiological situation. The stability is expressed as DT50 (in hours), representing the incubation time for 50 % of the complexes to decay.
After peptide incubation for 18 h at 37 °C the cells were either washed two times with PBS to remove free peptide and subsequently resuspended in 200 µl serum free RPMI medium or were incubated for 1 h at 37 °C with Brefeldin A (BrefA) at a final concentration of 10 µg/ml.
3. We agree with the reviewer that statistical analysis would be beneficial to support the results given in Fig. 2. We performed the statistical analysis and amended the respective text in the manuscript. (results, page 11, lines 180-181; Material and Methods, page 26, lines 482-483). We now extended the legend of Fig. 2 for clarity.
The evidence that the VIL9 binds best to the CD94/NKG2A receptor is given in Fig. 2b.
We appreciate the reviewers request to explain the percentages in table 2 comprehensively. The table shows the data from Fig. 3, however the data have been calculated using peptide empty T2E cells, the respective pT2E value has been subtracted. We specified that in the result section (page 12-13, lines 207-209). For clarity we completed the legend of table 2.
Minor concerns
1. We agree with the reviewer, the manuscript has been critically reviewed by a native speaker.
2. We agree with the reviewer and amended the scale of figure 1 accordingly.
3. We acknowledge the reviewers concern, the information of the BrefA experiments is given in the result section (page 9, 1st paragraph).
4. We agree with the reviewer and magnified the numbers in figure 4.

Reviewer 2 Report
The abstract should define an “artificial” infection…the ms actually doesn’t shed light on this
The sentence in line 72 needs re-writing
Try to reduce the abbreviations….PBR, PLC etc…they make it hard to read this work
I don’t understand the assays of HLA-E stabilization (Table 1, Fig 1), eg …
1. Is the sequence identical in HLA-C and UL40
2. The self citation (ref 30) does not explain the origin of the peptides tested
3. The legend reads “The….MFI value at t = 0 h was set as 100% to calculate DT50 values” so how did you obtain a MFI of ~140…what are the units?
4. I cant see how you obtained the TD50 values in table 2 given that the values with and without brefeldin appear identical for all peptides except VIL9
Table 2 is cited on line 103 but has no legend. What are #1 and #2? What are the units for the T-cell columns? Why are the columns beside SY10 and VIL9 shown as “-“?
Figures 2, 3 and 4 need clearer legends and show no clear pattern. Eg: In Fig 2, DQ13 and LIL9 are the optimal binders…but they don’t have optimal TD50 (Table 2) or show optimal T-cell activation (Fig 3)…again what are #1 and #2?
It would help if Fig 3 and 4a were aligned…so I could see if the same peptides activate T-cells
I don’t see the message of Fig 4b.
The manuscript should be shorter and clearer with a focus on the data presented
Author Response
Reviewer 2:
We agree with reviewer and amended the abstract.
We agree with reviewer and rewrote the sentence in line 72 (Introduction, page 7, 1st paragraph, lines 97-100).
We agree with the reviewer and reduced the abbreviations.
To ensure that the pHLA-E complexes are stable through the time of the functional assays, we determined the decay of the complexes. The stability of cell surface p:HLA-E complexes was monitored with inhibition of intracellular protein transport by BrefA that provides specific decay of p:HLA-E complexes on the cell surface at the initial time point. Furthermore, p:HLA-E complex stability was determined without inhibition of intracellular protein transport that reflects the physiological situation. The stability is expressed as DT50 (in hours), representing the incubation time for 50 % of the complexes to decay.
The peptide sequence VMAPRTLIL is identical in HLA-C and UL40. Usually the signal sequence of HLA-C is presented by HLA-E. However, in CMV infected cells HLA-Ia molecules are down regulated, while the viral gpUL40 that has the same sequence than the HLA-C signal peptide will be presented by HLA-E to prevent NK cells from recognition of the infected, HLA Ia-empty cells.
For clarity we amended the caption of table 1 (page 9).
We acknowledge the reviewers concern and extended the legend of table 2 (page 11, lines 164-165); the legend of figure 1 has been amended.
The DT50 values in table 2 represent the incubation time for 50 % of the complexes (based on the MFI at 0 h) to decay in the presence of Brefeldin A. We amended the text and the legend of table 2 for clarity (page 11, lines 164-165; results, page 9, 1st paragraph, lines 133-134).
According to the reviewers request we completed the legend of table 2 (page 11, lines 160-165).
We agree with the reviewer and completed the legends of figures 2, 3 and 4; the result section has been amended accordingly (page 11, 12, lines 180-182, 204). Figure 2 shows the amount of bound peptide at the start point (after 2h incubation) of the sNKG2/CD94 binding assay. Table 2 gives the data of the stability assays over time to determine how long the pHLA complexes were stable. The stability of the complexes had no influence on T cell activation. The legend of Table 2 has been extended and completed for clarity (page 11).
We acknowledge the reviewers request to see the alignment of Fig 3 and 4a. However, both figures answer different questions. The reason for the chosen representation was to first determine the peptides that trigger proliferation and then we analyzed the phenotypes of the respective peptides.
The message of figure 4b where we determined the amount of NKG2A/2C/CD94 on T cells pre- and post-stimulation has been extensively discussed in the discussion section (page 20).

Round 2
Reviewer 1 Report
The statistical analyses performed are not included as part of the Figures of the amended manuscript.
Author Response
We apologize for the confusion and now included the statistics in Figure 2.

Reviewer 2 Report
The paper was improved in the first revision but hard decisions are needed.
This paper is interesting but the data are really hard to follow and may be over-interpreted in some instances. The authors should
1. arrange for the manuscript to be read by a colleague and fix all the places where their colleague was unable to follow what was done. This should include explanation of abbreviations (eg: p:HLA-E, Bref A) the first time they are used and in the abbreviations list
2. re-write the legends of all figures so they make sense independent of the text.
3. Re arrange Figure 1 …., I cannot compare with and without Bref 4 when they are presented above each other. Perhaps a more important graph would be the difference between the two.
I suspect SY10 and VIL9 may be mislabeled as SY10 has better binding without Bref A…see line 113
I cant derive DT50 values from the graphs… this should be possible
4. In Table 2…the DT50 values of the peptides are really all the same…surely that would make the interpretation of the data easier. It would also mean that Table 2 wasn't needed as the message seems to be only that the long peptides stimulate proliferation and the short ones don't….the Discussion assumes they all have similar binding
5.The legend of Figure 2 suggests that significant differences are marked with an asterisk * but there is no * on either graph. And…how was the exp used to generate Fig 2b performed. You have not demonstrated that 3D12 cells don't express NKG2/CD94.
6. Fig 3 and the associated text explain the proliferation assay nicely. Is this the same assay that was used without explanation) in Table 2? The conclusion is effectively the same, long peptides stimulate proliferation
Author Response
We agree with the reviewer and let two colleagues critically read and correct the paper where the message was not transparent. A native speaker reviewed and corrected the English. As suggested by the reviewer we extended the abbreviation list with all abbreviations used in the manuscript.
As requested by the reviewer we rewrote the figure legends for clarity.
As requested by the reviewer we rearranged Figure 1 for clarity. Each peptide is now given in a separate graph showing results with and without BrefA for better comparison of the differences between the two. SY10 and VIL9 can be now better distinguished, VIL9 shows the best binding. We now included the DT50 values in the graphs. The dashed lines indicate the MFI after 50% of the complexes decayed. The crossing of a curve with this line marks the DT50 (reported in table 2).
We now included for clarity the DT50 values also in Figure 1, the differences between the peptides are now easier to interpret. We explained the origin of the data in the legend of Table 2.
We apologize for the confusion and now included the statistics in Figure 2. The designation “3D12” indicates the anti HLA-E specific antibody (Material and Methods, page 13, line 382. Using this antibody we identified pHLA-E complexes on T2 cells. We did not verified NKG2/CD94 expression on T2 cells, since we used this lymphoblastic cell line as target cells and NKG2/CD94 expression was not expected.
We thank the reviewer for raising our attention to the misleading information in Figure 3. In Table 2 we show the results of T cell proliferation that were calculated by subtracting the proliferation rate without peptide (%T2E = control) from the proliferation rate with the respective peptide (%p:T2E) from Figure 3. We extended the legend of Table 2 accordingly for clarity.
